# Construction and validation of the area level deprivation index for health research: A methodological study based on Nepal Demographic and Health Survey

Ishor Sharma[1]*, M. Karen Campbell[1,2], Marnin J. Heisel[1,2], Yun-Hee Choi[1], Isaac N. Luginaah[3], Jason Mulimba Were[1], Juan Camilo Vargas Gonzalez[1], Saverio Stranges[1,2,3,4]

1 Department of Epidemiology and Biostatistics, Western University, London, Ontario, Canada, 2 Lawson Health Research Institute, London, Ontario, Canada, 3 Department of Geography, Western University, London, Ontario, Canada, 4 Department of Population Health, Luxembourg Institute of Health, Strassen, Luxembourg

* isharm9@uwo.ca

**Data Availability Statement:** DHS data is freely available to use with the permission form DHS

## Abstract

Area-level factors may partly explain the heterogeneity in risk factors and disease distribution. Yet, there are a limited number of studies that focus on the development and validation of the area level construct and are primarily from high-income countries. The main objective of the study is to provide a methodological approach to construct and validate the area level construct, the Area Level Deprivation Index in low resource setting. A total of 14652 individuals from 11,203 households within 383 clusters (or areas) were selected from 2016-Nepal Demographic and Health survey. The index development involved sequential steps that included identification and screening of variables, variable reduction and extraction of the factors, and assessment of reliability and validity. Variables that could explain the underlying latent structure of area-level deprivation were selected from the dataset. These variables included: housing structure, household assets, and availability and accessibility of physical infrastructures such as roads, health care facilities, nearby towns, and geographic terrain. Initially, 26-variables were selected for the index development. A unifactorial model with 15-variables had the best fit to represent the underlying structure for area-level deprivation evidencing strong internal consistency (Cronbach's alpha = 0.93). Standardized scores for index ranged from 58.0 to 140.0, with higher scores signifying greater area-level deprivation. The newly constructed index showed relatively strong criterion validity with multi-dimensional poverty index (Pearson's correlation coefficient = 0.77) and relatively strong construct validity (Comparative Fit Index = 0.96; Tucker-Lewis Index = 0.94; standardized root mean square residual = 0.05; Root mean square error of approximation = 0.079). The factor structure was relatively consistent across different administrative regions. Area level deprivation index was constructed, and its validity and reliability was assessed. The index provides an opportunity to explore the area-level influence on disease outcome and health disparity.

program. https://dhsprogram.com/data/available-datasets.cfm.

**Funding:** The author(s) received no specific funding for this work.

**Competing interests:** The authors have declared that no competing interests exist.

## Introduction

Understanding how an area-level construct is associated with disease etiology is of recognized importance in the field of epidemiology and health research. In contrast to individual level research, area-level research focuses on the wider social and environmental contexts where an individual reside [1]. Inferences based on single area-level features such as literacy rates, median income, and unemployment rates are likely to provide an incomplete picture of the area's socio-demographic/economic/ecological context [2]. Moreover, high correlations between these variables make it a challenge to interpret the findings of the study [3]. Study by Townsend, suggested a composite measure based on the material (e.g., home and car owner-ship) and social features (e.g., unemployment rates and household crowding) to characterize the area level construct that is less likely to be influenced by changes in a single variable and thus better at capturing the underlying construct [4]. The area-level deprivation (AD) is a composite measure of area level construct and is defined as the "relative disadvantage an individual or a social group experiences in terms of access and control over economic, material or social resources and opportunities" [5].

Individuals from socially and economically deprived areas are often at an elevated risk of disease and negative health consequences, such as adverse birth outcomes, maternal mortality and morbidity, chronic conditions such as diabetes, hypertension, and mental health. Behavioral risk factors such as gambling, drug abuse, alcoholism, smoking, and inter-partner violence are relatively higher in such areas [6]. Similarly, AD is correlated with poor access to health services, higher levels of food insecurity, health-promoting behaviors, poorly built environments such as parks, walking space, and increased exposure to environmental pollutants [6–10]. This indicates area level construct as a significant determinant of the health and signifies its importance in the health research. The Townsend index, the Carstairs Index, and the Canadian Index of multiple deprivations are commonly used indices to assess AD [11–13]. Although, these indices are limited to generalizability and varies on their methodological approaches, used indicators such as poverty, racial/ethnic composition, illiteracy, unemployment, and housing characteristics are consistent across studies. However, the definition of these indicators varies [11–13]. Although, area-specific spatiotemporal components are recommended for assessing the area level deprivation, this component is largely neglected in assessing the area level deprivation [14].

Demographic and health surveys (DHS) are regularly conducted in most of the Low- and Middle-Income Countries (LMICs), primarily focused on maternal and child health However, with the changing disease patterns, many countries have also started collecting data on non communicable disease risk factors and chronic conditions. Cluster/Area that represents approximately 300 households are one of the sampling units (PSU) in DHS [15]. There are no specific measures that could characterize these area/clusters, limiting an opportunity to explore the area level variations in health disparity and disease outcomes. This could be an opportunity to pinpoint policymakers to identify hotspots of diseases or health status. The multidimensional poverty index (MPI) and the wealth index are often aggregated at the cluster level [16–18]. These indices are based on household indicators and do not incorporate social, and area-level spatial components such as access to health facilities, residential proximity to cities, and major roadways which could have a significant role in assessing underlying AD [19,20]. This emphasizes the need for an exhaustive, valid, and reliable area-level deprivation index (ADI) to better characterize the underlying area level construct.

A valid and reliable tool to measure AD could help to explore the disease etiology, identify high risk populations, and thus inform policy development and implementation. Using the

latest 2016 Nepal DHS, this paper outlines a systematic approach for the development and validation of the ADI.

## Materials and methods

Briefly, the Nepal DHS uses multistage stratified random sampling. In rural, wards were selected as PSU while in the urban regions one enumeration area (EA) was selected from each ward. From each PSU or EU approximately 30 households were selected for the survey [15]. Each PSU or EA was treated as an area or a cluster for index development.

### Process of index development

Index development follows previous methodological works and approaches [3,21–24]. Briefly, the steps involved i) selection of relevant variables, ii) screening and assessment of variables, iii) variable reduction and extraction of the factors, and iv) assessment of validity and reliability.

**Selection of the observed variables.** Variable selection was guided by the earlier studies; (primarily includes material, social and geographical features) [11–13,19,20], availability of the variables in the DHS dataset, and expert opinions. Based on these, a total of 26 aggregated and non-aggregated observed variables were selected which could explain the underlying construct; the Area-level deprivation. Aggregated variables were proportion or the average of the individuals/household's characteristics at the area-level. These variables falls under six domains: ethic heterogeneity, education, employment, household assets, household structure, access to public and social infrastructures. A detailed list of the variables is provided in appendix (S1 Table in S1 File).

**Variables screening and assessment.** Index construction might be inappropriate if observed variables are poorly correlated [25]. Variables with a low correlation (±0.30) in the correlation matrix and uniformly distributed across the clusters would be excluded. Uniform distribution means variables that are consistently present in almost same frequency in more than 90% of the clusters. The Kaiser-Meyer-Olkin (KMO) measure of sampling adequacy and Bartlett's test of sphericity were used to assess the suitability of data for index development [22,26]. The KMO indicates the proportion of variance caused by the underlying factors and ranges between 0 and 1, value above $\geq 0.7$ is suggested [26]. A significant Bartlett's test suggests the correlation matrix of variables are significantly different from the identity matrix indicating the chosen variables are suitable for data reduction [22].

**Variable reduction and factor extraction.** Principal component analysis (PCA) and Exploratory Factor Analysis (EFA) are commonly used techniques in construction of indices [3]. Before beginning with the selection of these techniques its better to understand how these two approaches work. In general, the total variance can be partitioned into common and unique variance. Common variance (or shared variance) is amount of variance that is shared amongst observed variables. Highly correlated variables share a lot of variances. Unique variance is any portion of total variance that is not common and could be either specific variance i.e., specific to the observed variable, or error variance which is basically a measurement error. PCA assumes the common variance takes total variance (common/shared and unique) whereas, EFA utilizes only common variance [27]. PCA maximizes the total variance of variables by integrating them into the weighted linear combinations to identify uncorrelated components. Where as, EFA explores the pattern of relationship amongst variables and their shared variance to create a latent construct [28]. As we are interested in the measurement of underlying latent construct; the area level deprivation, we selected EFA over PCA.

EFA was conducted using iterated principal factor estimation due to lack of multivariate normality [29]. The number of factors extracted were based on eigenvalues > 1, visual inspection of the scree plot, and the conceptual meaning of the factor [30]. Promax rotation was selected to assess factor loadings as the researcher expects some correlation between factors. Factor loading refers to the correlation of the variable with the latent structure. A more relaxed a priori criteria i.e., factor loadings <0.20 was used to exclude any variables from the factor [31]. A minimum of 3 variables with high loading is suggested for factor [30]. Communality is the proportion of each variable variance that is explained by the underlying factor. High factor loadings and communality suggest a stronger relation between variables and the latent factor. The variables were then weighted by their factor loading.

**Assessing index quality: Validity and reliability.** Validity of the index was assessed by content, construct, and criterion validity. Content validity refers to the accurate representation of the underlying construct; the study especially sought to reflect the full scope of the area-level deprivation construct with the ADI. Confirmatory factor analysis (CFA) was conducted to test the construct validity of the index based on the factors obtained from EFA in the randomly sample (n = 200). Maximum likelihood estimator with robust standard error was used. Root mean square error of approximation (RMSEA< 0.80), Comparative Fit Index (CFI ≥0.90) and the Tucker-Lewis Index (TLI ≥0.95) were used to assess model fit [32]. Modification indices was explored to identify model misfit areas. We also assessed for the place-based stability of the deprivation index (Invariance testing) by assessing factor loadings and its magnitude across three administrative regions (cluster, districts and ecological regions) [21]. Criterion validity was assessed by correlating the newly constructed deprivation index with the 2018-MPI [16]. Reliability was assessed using Cronbach's alpha. Anything above 0.7 was considered an indication of good internal consistency (internal reliability) [33].

Exploratory factor analysis was conducted using SAS version 9.4 (SAS Institute, Cary, NC, US), whereas confirmatory factor analysis was conducted in Mplus (Version 7, Muthén & Muthén, Los Angeles, CA, 2017). Spatial plotting was done using ArcGis-10.7 using the available cluster level geographic coordinate systems obtained from the Nepal DHS -2016 spatial data respiratory.

## Results

### Characteristics of the study population

In total, 14652 individuals from 11,203 households (99%) within 383 clusters were successfully interviewed. Numbers of houses in a cluster ranged from 10 to 74 (average in each cluster = 40). The mean age ± standard deviation (SD) of the sample was 38.61±17.60 years. More than half (57%) were female.

### Selection of the observed variables

Of the 26 variables, ethnic heterogeneity and proportion of the dependent aged population were almost consistent across the clusters. Most clusters (>90%) had a very high proportion (>95%) of households with a toilet, drinking water, a radio, and a separate kitchen and a very low proportion (<5%) owned a car or a shared washroom. These eight variables also showed poor correlations in the correlation matrix and hence were excluded for further analysis. Both the KMO (0.91) and the Bartlett's test of sphericity (p-value <0.001) suggested the appropriateness of remaining 18 variables for factor analysis.

**Table 1. Factor loading and shared variance (communality) for 15-variables at three different geographic levels (Cluster/area, districts, and sub-regional levels).**

| Area level variables | Region | | | |
|---|---|---|---|---|
| | Cluster | | District | Sub-region |
| | Factor loading | Communality | Factor loading | Factor loading |
| Proportion of households with illiterate population | 0.71 | 0.50 | 0.57 | 0.28 |
| Proportion of eligible population employed | 0.69 | 0.48 | 0.63 | 0.79 |
| Proportion of households with electricity | 0.46 | 0.21 | 0.46 | 0.71 |
| Proportion of households not exposed to mass media | 0.72 | 0.52 | 0.78 | 0.87 |
| Proportion of households without TV | 0.81 | 0.65 | 0.86 | 0.93 |
| Proportion of households without refrigerator | 0.85 | 0.72 | 0.88 | 0.95 |
| Proportion of households without motorcycle | 0.76 | 0.58 | 0.84 | 0.88 |
| Proportion of households with rudimentary floor | 0.91 | 0.82 | 0.93 | 0.94 |
| Proportion of households with rudimentary wall | 0.84 | 0.71 | 0.85 | 0.91 |
| Proportion of households without telephone | 0.61 | 0.37 | 0.64 | 0.61 |
| Proportion of households without clean energy source | 0.89 | 0.79 | 0.90 | 0.95 |
| Proportion of households without bank account | 0.63 | 0.40 | 0.72 | 0.91 |
| Proportion of households without soap for handwash | 0.71 | 0.50 | 0.73 | 0.75 |
| Average time required to reach nearby motorable road | 0.57 | 0.33 | 0.75 | 0.89 |
| Average time to reach to collect water | 0.48 | 0.23 | 0.45 | 0.63 |
| *Cronbach's alpha reliability* | 93.0 | | 0.94 | 0.97 |

## Variable reduction and factor extraction

An initial EFA with 18-variable revealed two factors, explaining almost 92% of variability (75% and 17% respectively). The number of factors was based on eigenvalues > 1 and the inspection of scree plot S1 Fig in S1 File. Using the Promax rotation, 15 variables loaded onto the first factor. Three variables (Altitude in meters, time to health facilities (in minutes), and proportion of household with bicycle) had factor loadings <0.20 on the first factor. The second factor with these three variables loaded on it did not provide any specific theoretical explanation on a grouping of those variables. Hence, a unidimensional 15-variable ADI was proposed. S2 Table in S1 File provides the descriptive statistics for the variables selected for the final analysis.

Following identification of the variables for inclusion in the ADI, an EFA was re-rerun, with the 15-variables forced onto a single latent factor. The resulting factor loadings ranged from 0.91 for households with a rudimentary floor structure to 0.46 for households with electricity (Table 1). Communalities were strong (>0.5) for the majority (60%) of the variables. The 15 variables were then weighted by their factor loading. The resulting ADI was then standardized to a mean = 100 and SD = 20.

## Area-level deprivation index

The standardized ADI thus ranges from a minimum of 58.0 to a maximum of 140.0, with higher scores representing a higher level of deprivation. (Fig 1) shows the distribution of the ADI across the geographic regions in Nepal. Dark shaded areas in the background shows higher level of deprivation in contrast, lighter regions shows lower deprivation.

## Assessing Index quality: Validity and reliability

Content and face validity: The conceptual framework for the Nepal area ADI was primarily based on the framework of Messer and Townsend that includes material and social constructs.

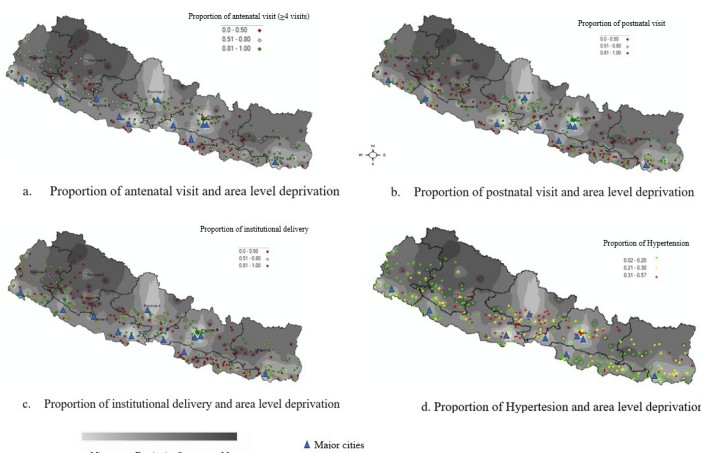

**Fig 1. Area level deprivation and a) Proportion of antenatal visit (visit <4); b) Proportion of postnatal visit; c) Proportion of institutional delivery; d) Proportion of hypertensive individuals across the clusters.** Shapefile republished from DIVA-GIS database (https://www.diva-gis.org/) under a CC BY license, with permission from Global Administrative Areas (GADM), original copyright 2018. DIVA-GIS is a free and open-source geographic information system (GIS) to make maps of species distribution data and analyze these data. Data were provided by the demographic and health survey, Maps created in ArcGIS 10.7.

[1,5] As these frameworks may not capture multiple aspects of area level deprivation other variables were also added based on literature reviews, availability of variables on dataset, and from experts' opinions.

The North-Eastern and North-Western regions of the country appear to have a higher deprivation (Fig 1). The deprivation is in line with the urbanization, as urbanized areas appear to have a lower deprivation as compared to rural. We assessed if the correlation (Pearson's Correlation (r)) between AD and health and health services utilization indicators are in line with the published studies. The proportion of Institutional delivery (r = -0.64), the proportion of hypertension (r = -0.32), and the proportion of obesity (r = -0.42) at the area level were negatively correlated with the deprivation score. Similarly, average time to reach nearby health facilities (r = 0.30) and the proportion of ANC visit (<4 visits) (r = 0.47) had a significant positive correlation with the deprivation score.

Criterion Validity: The newly constructed ADI was strongly correlated with the latest 2018 Multidimensional Poverty Index of Nepal (r = 0.77).

Construct validity: The single factor model without correlated error had a poor fit (RMSEA = 0.14, CFI = 0.88, TLI = 0.85), suggesting that the latent construct might be missing important relationships. As suggested by modification index and theoretical reasoning, the re-specified model S2 Fig in S1 File (Appendix) with correlated residuals showed a substantial improvement RMSEA = 0.079, CFI = 0.96, and TLI = 0.94. The 15-variables significantly loaded on the single latent factor with a standardized path coefficient ranging from a minimum of 0.38 to a maximum of 0.92. Proportion of households with no electricity and average time to collect drinking water showed poor convergent validity (β<0.5).

The results of the factorial invariance suggested that factor loading for 15-variables remained similar (Table 1). The reliability coefficient was similar, and the percentage of the variance explained in all three cases were more than 55%.

Reliability as assessed by the Cronbach coefficient was 93.0, showing strong internal consistency.

## Discussion

The current study developed and assessed the validity and reliability of ADI based on the Nepal DHS. The standardized Nepal ADI ranged from 58 to 140, with higher scores representing greater area-level deprivation. The extent of deprivation seemed to vary significantly within and across the geographic areas. Despite the wide use of ADI in different settings, questions abound with regards to their validity and reliability [34]. The current study did not only focus on the index development but also provides quality assessment using three criteria: validity, reliability, and responsiveness.

Conventionally, deprivation indices have focused primarily on material well-being.[4] However, with emerging multidimensional outlook of health, the deprivation characterization has been extended to incorporate broader domains [14]. In the current study, we used multiple domains to capture various aspects of AD to assure the content validity [3,4,21–24]. The material domain included household structure and assets. Social domains include literacy levels, disadvantaged population, and dependent population. Geospatial domains include access to nearby cities, access to roads and health infrastructures. There is strong theoretical background for the inclusion of these variables. For example, unemployment reflects finance/income which impact on accessing healthy foods and health care, illiteracy predisposes individual's poor understanding of the public health information, better job opportunity, etc. Household structure and assets are the proxy for economic status. Geographical difficulty in the context of Nepal might limit individuals from accessing health services, access to roads and, food commodities. All these observed variables could reflect the underlying latent structure of deprivation. Despite including these variables, the index was still limited by the exclusion of important variables such as income, availability and accessibility of educational institutions, type of motorable roads, which may have a significant role of defining AD. Eight variables were excluded in the initial step. The poorly correlated and homogenously distributed variables are unlikely to explain the underlying construct as they may not explain substantial variations [25].

A 15-variable unidimensional model without correlated error terms showed a poor construct validity. Because the current model contained only a single factor, additional paths could only be added as correlated residuals [35]. All these correlated residuals have a content relationship. All the 15 variables in re-specified model loaded significantly on a single latent factor. Criterion validity was established based on strong correlation between ADI and 2018 MPI. MPI resembles the content validity of the ADI to some extent. The minimal discrepancy is probably due to the inclusion of different observed variables; MPI includes the components of nutrition and child mortality as one of the dimensions which is not included in ADI [16]. Similarly, ADI includes variables such as distance to health facilities, cities, access to roads which are not in MPI.

ADI showed a strong internal consistency. The factor loadings for variables were relatively consistent at various geographic levels, reflecting that the index measured same latent construct in different context [21]. Although the internal consistency and factor loadings varied slightly across the areas, the one factor model was relatively persistent.

Responsiveness relates to the ability of the index to capture variations across time, place, and person [36]. The ADI showed variations across the geographic areas. These variations partly could explain the heterogeneity in health outcomes and the risk factors across the geographic region. As seen in Fig 1, The ADI showed significant correlation with the poor utilization of health services such as the antenatal care utilization, and institutional deliveries. These associations are plausible and consistent across the study. Deprived areas lack basic health infrastructures, and even if present, there is lower awareness thereby resulting to poor health-seeking behaviors leading to poor health outcomes [7]. Further, the deprived areas seemed to

be associated with a lower risk of hypertension and overweight/obesity. This clearly reflects that the area deprivation might be associated with food accessibility, hunger, labour intense society in the context of Nepal. In contrast, less deprived areas might have better access to refined and processed foods, more sedentary behaviors and lifestyles which could lead to the elevated risk of chronic conditions such as hypertension and overweight/obesity [8,37].

The index development process is some what constricted towards the unidimensional construct. A concept of 'essential unidimensionality' where a secondary minor latent variable is often possible [38]. Initial EFA revealed two underlying factors. Two of the three variable (time to health facility and altitude) loaded strongly on the second factor might be associated with geographical difficulty. There is always possibility of having multiple underlying constructs for AD if we had more observed variables for example having more on representation of the spatial variables such as weather pattern. From the theoretical point of view AD could be associated with education, occupation, household structure, household assets, place of residence which makes logical reason to put them together in the unidimensional construct, despite they may be a reflective variable for another minor latent construct. We found that the single aggregate index is conceptually convenient for interpretation, because of which we used a lenient approach to include the observed variables using lower factor loadings [38,39]. The unidimensional construct for AD seemed to be conceptually and empirically valid.

As this study was based on the Nepalese context, the index is limited in generalizability. Area-level characterization of the scales developed using data driven techniques are sensitive to space and time [14]. However, as highlighted earlier one of the main propose of the study was also to provide an outline for construction and validation of area-level composite measure; focused on DHS with necessary country specific modification in the observed variables.

## Conclusion

The 15-item ADI was constructed based on Nepal DHS and was assessed for its validity and reliability. The strong validity and reliability suggests its applicability in health research to explore area level disparity in health and disease outcomes. The index showed relatively strong criterion validity with multi-dimensional poverty index and relatively strong construct validity. The factor structure was relatively consistent across different administrative regions. Content validity was assured using the framework by Messer and Townsend and literature review. Face validity assessed with health indicators across the geographic regions using the correlation coefficients. High Cronbach's alpha coefficient showed relatively strong internal consistency.

## Supporting information

**S1 File. S1 File contains supporting tables and supporting figures.**
(PDF)

## Acknowledgments

We like to thank participants of the survey as well to DHS program for providing the data for Nepal DHS-2016. Authors thank Ontario Trillium Foundation for the PhD fellowship.

## Author Contributions

**Conceptualization:** Ishor Sharma, M. Karen Campbell, Marnin J. Heisel, Jason Mulimba Were, Juan Camilo Vargas Gonzalez, Saverio Stranges.

**Data curation:** Ishor Sharma.

**Formal analysis:** Ishor Sharma, Marnin J. Heisel.

**Methodology:** Ishor Sharma, M. Karen Campbell, Yun-Hee Choi, Isaac N. Luginaah, Jason Mulimba Were, Juan Camilo Vargas Gonzalez, Saverio Stranges.

**Software:** Ishor Sharma.

**Supervision:** M. Karen Campbell, Marnin J. Heisel, Yun-Hee Choi, Isaac N. Luginaah, Jason Mulimba Were, Saverio Stranges.

**Validation:** Ishor Sharma.

**Visualization:** Ishor Sharma.

**Writing – original draft:** Ishor Sharma.

**Writing – review & editing:** Ishor Sharma, M. Karen Campbell, Marnin J. Heisel, Yun-Hee Choi, Isaac N. Luginaah, Jason Mulimba Were, Juan Camilo Vargas Gonzalez, Saverio Stranges.

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
