## [Decision Letter · Decision Letter 0]

12 Apr 2023

PONE-D-22-24690Construction and validation of the area level deprivation index for health research: A methodological study based on Nepal demographic health surveyPLOS ONE

Dear Dr. Sharma,

Thank you for submitting your manuscript to PLOS ONE. After careful consideration, we feel that it has merit but does not fully meet PLOS ONE’s publication criteria as it currently stands. Therefore, we invite you to submit a revised version of the manuscript that addresses the points raised during the review process.

We look forward to receiving your revised manuscript.

Kind regards,

Omid Dadras, MD, PhD

Academic Editor

PLOS ONE

Journal Requirements:

2. PLOS requires an ORCID iD for the corresponding author in Editorial Manager on papers submitted after December 6th, 2016. Please ensure that you have an ORCID iD and that it is validated in Editorial Manager. To do this, go to ‘Update my Information’ (in the upper left-hand corner of the main menu), and click on the Fetch/Validate link next to the ORCID field. This will take you to the ORCID site and allow you to create a new iD or authenticate a pre-existing iD in Editorial Manager. Please see the following video for instructions on linking an ORCID iD to your Editorial Manager account: https://www.youtube.com/watch?v=_xcclfuvtxQ.

Reviewers' comments:

Reviewer's Responses to Questions

**Comments to the Author**

1. Is the manuscript technically sound, and do the data support the conclusions?

Reviewer #1: Yes

Reviewer #2: No

2. Has the statistical analysis been performed appropriately and rigorously? 

Reviewer #1: Yes

Reviewer #2: N/A

3. Have the authors made all data underlying the findings in their manuscript fully available?

Reviewer #1: Yes

Reviewer #2: No

4. Is the manuscript presented in an intelligible fashion and written in standard English?

Reviewer #1: Yes

Reviewer #2: No

5. Review Comments to the Author

Reviewer #1: The manuscript is well written piece of work and report construction of an important tool for use in health research. There are typos (e.g page 5, first paragraph line 2: ....a total of 26 aggregated and non-aggregated observed were selected that could explain.....) that need to be corrected.

Reviewer #2: 1. Construction and validation of the area level deprivation index for health research: A methodological study based on Nepal demographic health survey

Here, term "health research" used in topic is not explained under study. Also, it has to be Nepal demographic and health survey, 2016. So, Research topic need to be rephrased.

3. Author citation need to be precised; there is repetition of similar working positions.

4. Abstract didn't match body of study. Please follow submission guidelines provided in Journal.

5. In abstract part; Objective and conclusion is clearly missing.

6. In abstract, Most of the sentences is written wrongly.

Example 1; This paper provides a methodological approach to construct and validate the area level construct, the Area Level Deprivation Index especially in low resource setting.

Such sentences don't give any clear meaning.

Example 2; Data was based on secondary data from 2016-Nepal Demographic Health Survey.

Please write the sentence accurately.

7. How construction and validation of the area level deprivation index is done within study? Abstract is not explaining on it.

8. Put Reference before full stop mark.

10. Introduction of study has to be written placing statements in a logical sequence.

11. The definition of area-level deprivation is explained only at the last part (reference 5). Actually, introduction part has to be initiated by explaining clear concept on area level deprivation.

12. Also, there is common mistake like Townsend, suggested a composite measure based on the material (e.g.,

home and car ownership) and social features....

- It is to be written like Study done by Townsend suggests that.....

13. Details on the 2016 Nepal DHS can be found elsewhere [15]. Please clarify elsewhere??

14. Briefly, the Nepal DHS uses multistage stratified random sampling. Explain appropriate research materials and methods used in your study, not in DHS of Nepal.

- Methodological details need to be be rewritten in excellent way.

15. Index development follows previous methodological works and approaches. [3,21-24]. Write how present study methods work in your study.

16. Variable selection was guided by the earlier studies, [11-13,19,20]. Not explained clearly again?

17.In result part, clarify the specific scale used in this study for validation of area level deprivation index. Table and figure is explained differently (not found in order). This is fruitless way to represent your result.

18. Discussion part is not explained correctly. Here, compare your results with other studies and explain reasons for any discrepancies and similarities.

19. Conclusion failed to explain how 15-variable ADI shows strong validity and reliability in this study??

20. Acknowledgement has to be done properly too.

6. PLOS authors have the option to publish the peer review history of their article (what does this mean?). If published, this will include your full peer review and any attached files.

Reviewer #1: **Yes: **Professor Abdolreza Shaghaghi

Reviewer #2: No

---

## [Author Response · Author response to Decision Letter 0]

17 Sep 2023

Academic Editor

PLOS requires an ORCID iD for the corresponding author in Editorial Manager on papers submitted after December 6th, 2016. Please ensure that you have an ORCID iD and that it is validated in Editorial Manager. To do this, go to ‘Update my Information’ (in the upper left-hand corner of the main menu), and click on the Fetch/Validate link next to the ORCID field. This will take you to the ORCID site and allow you to create a new iD or authenticate a pre-existing iD in Editorial Manager. Please see the following video for instructions on linking an ORCID iD to your Editorial Manager account: https://www.youtube.com/watch?v=_xcclfuvtxQ.

Response: Orchid ID is validated in the Editorial Manager

We note that you have stated that you will provide repository information for your data at acceptance. Should your manuscript be accepted for publication, we will hold it until you provide the relevant accession numbers or DOIs necessary to access your data. If you wish to make changes to your Data Availability statement, please describe these changes in your cover letter and we will update your Data Availability statement to reflect the information you provide.

Response: Data used for the following study was provided by Demographic and Health Survey program which is freely accessible through the following link https://dhsprogram.com/

Your ethics statement should only appear in the Methods section of your manuscript. If your ethics statement is written in any section besides the Methods, please move it to the Methods section and delete it from any other section. Please ensure that your ethics statement is included in your manuscript, as the ethics statement entered the online submission form will not be published alongside your manuscript.

Response: Ethics statement is provided in the method section as

Ethics approval We used secondary anonymous data, hence, obtaining ethical approval was not needed for this study. However, we asked permission to use the data files from DHS Program. The DHS was conduced after the ethical approval from Nepal Health Research Council (NHRC) Reg No: 329/2015.

We note that Figure 1 in your submission contain [map/satellite] images which may be copyrighted. All PLOS content is published under the Creative Commons Attribution License (CC BY 4.0), which means that the manuscript, images, and Supporting Information files will be freely available online, and any third party is permitted to access, download, copy, distribute, and use these materials in any way, even commercially, with proper attribution. For these reasons, we cannot publish previously copyrighted maps or satellite images created using proprietary data, such as Google software (Google Maps, Street View, and Earth). For more information, see our copyright guidelines: http://journals.plos.org/plosone/s/licenses-and-copyright.

Response: Following sentence added

Shapefile republished from DIVA-GIS database (https://www.diva-gis.org/) under a CC BY license, with permission from Global Administrative Areas (GADM), original copyright 2018. DIVA-GIS is a free and open-source geographic information system (GIS) to make maps of species distribution data and analyze these data. Data were provided by the demographic and health survey, Maps created in ArcGIS 10.7. 

Reviewer 1

Reviewer #1: The manuscript is well written piece of work and report construction of an important tool for use in health research. There are typos (e.g page 5, first paragraph line 2: ....a total of 26 aggregated and non-aggregated observed were selected that could explain.....) that need to be corrected.

Response: The following sentence is revised as;

Variable selection was guided by the earlier studies (Material, social and geographical features), [11-13,19,20] availability of the variables in the DHS dataset, and expert opinions. Based on these, a total of 26 aggregated and non-aggregated observed variables were selected which could explain the underlying construct; the Area-level deprivation.

Additionally, the manuscript is reviewed for typos wherever applicable.

Reviewer 2

Reviewer #2: 1. Construction and validation of the area level deprivation index for health research: A methodological study based on Nepal demographic health survey

Here, term "health research" used in topic is not explained under study. Also, it has to be Nepal demographic and health survey, 2016. So, Research topic need to be rephrased.

Response:

a. Nepal demographic health survey is rephrased to Nepal demographic and health survey.

b. Following sentence is revised to add the importance of area deprivation in the health research.

Individuals from socially and economically deprived areas are often at an elevated risk of disease and negative health consequences, such as adverse birth outcomes, maternal mortality and morbidity, chronic conditions such as diabetes, hypertension, and mental health. Behavioral risk factors such as gambling, drug abuse, alcoholism, smoking, and inter-partner violence are relatively higher in such areas [6]. Similarly, AD is correlated with poor access to health services, higher levels of food insecurity, health-promoting behaviors, poorly built environments such as parks, walking space, and increased exposure to environmental pollutants [6-10]. This indicates area level construct as a significant determinant of the health and signifies its importance in the health research. 

3. Author citation need to be precise; there is repetition of similar working positions.

Response: Addressed wherever applicable.

Abstract

Abstract didn't match body of study. Please follow submission guidelines provided in Journal.

5. In abstract part, Objective and conclusion is clearly missing.

6. In abstract, most of the sentences is written wrongly.

Example 1; This paper provides a methodological approach to construct and validate the area level construct, the Area Level Deprivation Index especially in low resource setting.

Such sentences don't give any clear meaning.

Example 2; Data was based on secondary data from 2016-Nepal Demographic Health Survey.

Please write the sentence accurately.

7. How construction and validation of the area level deprivation index is done within study? Abstract is not explaining on it.

Response: Abstract is revised to make the suggested changes.

Area-level factors may partly explain the heterogeneity in risk factors and disease distribution. Yet, there are a limited number of studies that focus on the development and validation of the area level construct and are primarily from high-income countries. The main objective of the study is to provide a methodological approach to construct and validate the area level construct, the Area Level Deprivation Index in low resource setting. 

A total of 14652 individuals from 11,203 households within 383 clusters (or areas) were selected from 2016-Nepal Demographic and Health survey. The index development involved sequential steps that included identification and screening of variables, variable reduction and extraction of the factors, and assessment of reliability and validity. Variables that could explain the underlying latent structure of area-level deprivation were selected from the dataset. These variables included: housing structure, household assets, and availability and accessibility of physical infrastructures such as roads, health care facilities, nearby towns, and geographic terrain. 

Initially, 26-variables were selected for the index development. A unifactorial model with 15-variables had the best fit to represent the underlying structure for area-level deprivation evidencing strong internal consistency (Cronbach’s alpha = 0.93). Standardized scores for index ranged from 58.0 to 140.0, with higher scores signifying greater area-level deprivation. The newly constructed index showed relatively strong criterion validity with multi-dimensional poverty index (Pearson’s correlation coefficient=0.77) and relatively strong construct validity (Comparative Fit Index = 0.96; Tucker-Lewis Index= 0.94; standardized root mean square residual = 0.05; Root mean square error of approximation= 0.079). The factor structure was relatively consistent across different administrative regions. 

Area level deprivation index was constructed, and its validity and reliability was assessed. The index provides an opportunity to explore the area-level influence on disease outcome and health disparity.

8. Put Reference before full stop mark.

Response: Changed

10. Introduction of study has to be written placing statements in a logical sequence.

Response: Edited wherever applicable.

11. The definition of area-level deprivation is explained only at the last part (reference 5). Actually, introduction part has to be initiated by explaining clear concept on area level deprivation.

Response: I agree, there is some discrepancy between Reviewer 2 and the authors regarding the writing format. We have revised the manuscript and necessary modification were done wherever applicable.

12. Also, there is common mistake like Townsend, suggested a composite measure based on the material (e.g., home and car ownership) and social features....

- It is to be written like Study done by Townsend suggests that.....

Response: Study by Townsend, suggested a composite measure based ………..

13. Details on the 2016 Nepal DHS can be found elsewhere [15]. Please clarify elsewhere??

14. Briefly, the Nepal DHS uses multistage stratified random sampling. Explain appropriate research materials and methods used in your study, not in DHS of Nepal. 

Response: The current manuscript has the word limit of 2000 which is relatively limited compared to other journals. A consequence of which we are providing the references to the published materials. However, we have tried to provide information as briefly as possible. Following section is added in the revised draft.

Briefly, the Nepal DHS uses multistage stratified random sampling. In rural, wards were selected as PSU while in the urban regions one enumeration area (EA) was selected from each ward. From each PSU or EU approximately 30 households were selected for the survey. [15]. Each PSU or EA was treated as an area or a cluster for index development. 

15. Index development follows previous methodological works and approaches. [3,21-24]. Write how present study methods work in your study.

- Methodological details need to be be rewritten in excellent way.

Response: The following sections: highlights the brief overview of steps in the current study

Index development follows previous methodological works and approaches [3,21-24]. Briefly, the steps involved i) selection of relevant variables, ii) screening and assessment of variables, iii) variable reduction and extraction of the factors, and iv) assessment of validity and reliability. 

Details on each step is provided in the subsequent sections.

16. Variable selection was guided by the earlier studies, [11-13,19,20]. Not explained clearly again?

Response: Due to the limited, words count in the manuscript we have tried to be as succinct as possible. The following sections is revised as

Variable selection was guided by the earlier studies; (primarily includes material, social and geographical features), [11-13,19,20] availability of the variables in the DHS dataset, and expert opinions.

17.In result part, clarify the specific scale used in this study for validation of area level deprivation index. Table and figure is explained differently (not found in order). This is fruitless way to represent your result.

Response: As a part of assessing the criterion validity we compared the newly constructed 15- item ADI with the latest 2018 Multidimensional Poverty Index of Nepal. The following section added in the introduction section to explain the scale. 

The multidimensional poverty index (MPI) and the wealth index are often aggregated at the cluster level [16,18]. These indices are based on household indicators and do not incorporate social, and area-level spatial components such as access to health facilities, residential proximity to cities, and major roadways which could have a significant role in assessing underlying AD [19,20].

I agree with the author, that tables and figures although in orders are not described sequentially in the manuscript. This is primarily due to the nature of the methodological paper we keep on referencing to the earlier figures and tables. However, I have tried my best to maintain the logical order. 

18. Discussion part is not explained correctly. Here, compare your results with other studies and explain reasons for any discrepancies and similarities.

Response: Due to the nature of the study, we are more focused on the internal validity of the scale. However, strengths and limitations of the other studies such as Townsend index, Carstairs index, Canadian index of multiple deprivation, multidimensional poverty index, etc. are described in the introduction section. 

19. Conclusion failed to explain how 15-variable ADI shows strong validity and reliability in this study??

Response: Conclusion revised as

The 15-item ADI was constructed based on Nepal DHS and was assessed for its validity and reliability. The strong validity and reliability suggests its applicability in health research to explore area level disparity in health and disease outcomes. The index showed relatively strong criterion validity with multi-dimensional poverty index and relatively strong construct validity. The factor structure was relatively consistent across different administrative regions. Content validity was assured using the framework by Messer and Townsend and literature review. Face validity assessed with health indicators across the geographic regions using the correlation coefficients. High Cronbach’s alpha coefficient showed relatively strong internal consistency.

20. Acknowledgement must be done properly too.

Response: We like to thank participants of the survey as well to DHS program for providing the data for Nepal DHS-2016. First author thank Ontario Trillium Foundation for the PhD fellowship.

---

## [Editor Report · Decision Letter 1]

16 Oct 2023

Construction and validation of the area level deprivation index for health research: A methodological study based on Nepal demographic and health survey

PONE-D-22-24690R1

Dear Dr. Ishor Sharma

We’re pleased to inform you that your manuscript has been judged scientifically suitable for publication and will be formally accepted for publication once it meets all outstanding technical requirements.

Kind regards,

Omid Dadras, MD, PhD

Academic Editor

PLOS ONE
---

## [Editor Report · Acceptance letter]

6 Nov 2023

PONE-D-22-24690R1 

Construction and validation of the area level deprivation index for health research: A methodological study based on Nepal demographic and health survey 

Dear Dr. Sharma:

I'm pleased to inform you that your manuscript has been deemed suitable for publication in PLOS ONE. Congratulations! Your manuscript is now with our production department. 

Kind regards, 

on behalf of

Dr Omid Dadras 

Academic Editor

PLOS ONE